# Detection of Biomarker Using Aptasensors to Determine the Type of Diabetes

**DOI:** 10.3390/diagnostics13122035

**Published:** 2023-06-12

**Authors:** Dinda Exelsa Mulyani, Iman Permana Maksum

**Affiliations:** Department of Chemistry, Faculty of Mathematics and Natural Sciences, Universitas Padjadjaran, Sumedang 45363, Indonesia; dinda19003@mail.unpad.ac.id

**Keywords:** aptasensor, aptamer, diabetes mellitus, biosensor

## Abstract

Diabetes mellitus (DM) is a metabolic disorder characterized by elevated blood glucose levels. This disease is so serious that many experts refer to it as the “silent killer”. The early detection of diabetes mellitus, whether type 1, type 2 or mitochondrial, is crucial because it can improve the success of treatment and the quality of life for patients. Aptamer-based biosensor diagnosis methods have been widely developed because they have high sensitivity and selectivity in detecting biomarkers of various diseases. Aptamers are short sequences of oligonucleotides or proteins that recognize specific ligands and bind to various target molecules, ranging from small ions to large proteins. They are promising diagnostic molecules due to their high sensitivity and selectivity, ease of modification, low toxicity, and high stability. This article aims to summarize the progress of detection methods, including detection principles, sensitivity, selectivity, and the performance of detection devices, to distinguish between types of diabetes mellitus using electrochemical aptasensors with biomarkers such as glucose, insulin, HbA1c, GHSA, and ATP.

## 1. Introduction

Diabetes mellitus (DM) is a metabolic disorder characterized by increased blood glucose levels (hyperglycemia) resulting from defects in insulin secretion and insulin action [1]. According to the International Diabetes Federation (IDF), in 2021, there were 537 million adults aged 20–79 years with diabetes worldwide, which means that 1 in 10 people suffer from this disease. Moreover, diabetes kills around 6.7 million people every year, which is equivalent to one person dying every 5 s [2]. Several biomarkers are used as tests for DM patients, such as oral glucose tolerance tests (OTGG), fasting glucose level, insulin levels, glycated hemoglobin (HbA1c), and glycated albumin (GHSA) levels [3,4]. Various techniques have been developed to detect diabetes, including the HPLC-UV technique [5], fluorescence [6,7,8,9], chemiluminescence [10], ion exchange chromatography [11], immunoassay [12], and capillary electrophoresis [13]. However, measuring these biomarkers with instruments is not suitable for on-site monitoring [13,14].

This article will discuss the detection of biomarkers using aptamer bioreceptors, commonly called aptasensors, to determine the type of diabetes in patients. Aptamers are short, single-stranded DNA/RNA molecules that can recognize specific target molecules. The characteristics of aptamers make them very suitable for use as bioreceptors; they have good affinity for biosensors and offer an alternative approach that involves the use of small DNA molecules that can bind to specific targets with a very high affinity and specificity, replacing antibodies [15].

## 2. Methods

The method used to obtain systematic reviews is by searching for keywords such as pathophysiology of diabetes, diabetes mellitus detection, diabetes mellitus biomarkers, aptasensors and biosensors. Furthermore, journal literature or previous research from Sciene Direct, PubMed and Google Scholar were searched. Articles that are relevant to the discussion of the review were selected.

## 3. Result

### 3.1. Diabetes Mellitus (DM)

Diabetes mellitus, commonly referred to as DM, is a metabolic disorder characterized by prolonged high blood sugar levels or hyperglycemia. This disease is very serious, which is why many experts call it the silent killer [16]. The main complications of diabetes are micro-vascular complications (retinopathy, nephropathy, neuropathy) and macro-vascular complications (ischemic heart disease, stroke, peripheral vascular disease) [3,17,18]. Diabetes can also occur during pregnancy and alongside a number of conditions, including genetic disorders, drug or chemical toxicity, endocrinopathies, insulin receptor disorders, and diseases of the exocrine pancreas [18,19,20]. There are two types of diabetes that are commonly known that can be seen in Figure 1.

Type 1 diabetes occurs due to an autoimmune process that causes damage to the pancreatic beta cells. This leads to a decrease in insulin production [21]. This disease is closely related to premature morbidity and mortality, and can lead to complications such as neuropathy, nephropathy, and retinopathy. Patients with this condition require lifelong insulin administration to prevent hyperglycemia and ketoacidosis [22]. The pathology of type 1 DM involves the process of damage to insulin-producing cells by the immune system, caused by internal (genetic) and external (environmental) factors. This damage means that the hormone insulin used to regulate the movement of glucose from the bloodstream into cells for use by the metabolism is not produced. In type 1 DM disease, an autoimmune process occurs in pancreatic beta cells, where inflammation occurs involving the proinflammatory cytokines IL-1, TNF-alpha and INF-gamma induced by T lymphocytes. These cytokines can activate apoptosis [23] so that the pancreas stops producing insulin and blood glucose levels increase [24,25]. Increased glucose levels can lead to the transportation of glucose into the urine (glucosuria), resulting in osmotic diusesis because of excessive urine output (polyurea). Increased glucose levels in DM sufferers can be used as a biomarker for the early detection of diabetes to prevent the occurrence of diabetes complications. Diabetes can also be caused by the glycation of several proteins, namely HbA1c and GHSA. Blood glucose criteria for the diagnosis of DM are as follows [26] (Table 1):

Type 2 diabetes is caused by permanent hyperinsulinemia due to decreased insulin secretion, insulin resistance, or both [27]. This disease is usually caused by factors such as age, obesity, and lifestyle [28]. The pathophysiology of type 2 diabetes involves a disturbance of glucose homeostasis. Under normal conditions, insulin is secreted by pancreatic beta cells when glucose levels increase, leading to glycolysis and inhibiting the hormone glucagon as a directive for gluconeogenesis, thereby suppressing glucose production in the liver and muscles [29,30,31]. In patients with type 2 DM, insulin cannot direct the opening of the “gates”, so glucose cannot enter the tissues because insulin resistance occurs, meaning that the insulin hormone is rejected by target cells, and the glucose metabolism does not function in cells [27]. In addition to insulin levels, there is a type of diabetes related to the respiratory chain, called mitochondrial diabetes, caused by the dysfunction of insulin secretion due to an inhibition of the production of adenosine triphosphate (ATP), which is needed for insulin secretion. This dysfunction is related to mutations in the DNA tRNA genes of mitochondria (mtDNA) [32] (Figure 2).

When blood glucose levels are high, the glucose metabolism is activated, producing ATP and increasing its concentration in pancreatic β cells. This, in turn, induces the closure of K^+^ channels in the plasma membrane, leading to membrane depolarization. In response to the membrane depolarization, Ca^2+^ channels open, allowing for Ca^2+^ to flow into the cell. When the concentration of Ca^2+^ in the cytosol is high enough, it can trigger the release of the insulin hormone through exocytosis. In mitochondrial diabetes, ATP deficiency occurs, which inhibits insulin secretion and results in hyperglycemia. As a result, glucose in the blood cannot be taken up by gluscose transporter (GLUT) and remains in the blood [34] (Figure 3).

### 3.2. Aptasensor

In 1962, Calrk and Lysons first used the term biosensor [36]. A biosensor is a device capable of providing quantitative or semi-quantitative analytical information that combines the specificity of biological recognition mechanisms with physical transduction techniques that can be based on electrochemical, mechanical, thermal, or optical sensing principles [36,37]. Biosensors have many operational advantages, namely fast detection, ease of use, high sensitivity, cost effectiveness, and ease of mass production [38]. One type of biosensor is an aptasensor. This aptasensor uses the aptamer as its bioreceptor.

The aptasensor must have four main component parts [39], namely sensitive elements (bioreceptors), transducers, detection systems, and transporters. Bioreceptors are binding/interaction media between a target and a biosensor device. The transducer functions to change the interactions that occur in the bioreceptor so that they can be read, such as nanoparticles, nanoporous, nanowires, and fluorophores, which are directly proportional to the number of molecules that are bound or reacting to the surface of the sensor that can be connected to the reader [40]. The detection system measures the signal from the interaction that occurs. Lastly, there are transporters that function to integrate the steps of a process.

### 3.3. Aptamer

Aptamers are short sequences ranging from 40 to 180 nucleotides or peptides with 10–30 amino acid residues that recognize specific ligands and bind to a wide range of target molecules ranging from small ions to large proteins with high affinity and specificity [41]. The term aptamer comes from the word ‘aptus’, which means ‘fit’ [42]. Aptamers can be either single-strandded DNA (ssDNA) or RNA, with the only difference being the nitrogenous bases, namely thymine and uracil. DNA and RNA aptamers have the same specificity and affinity for targets; the difference is that DNA aptamers are more stable, while RNA aptamers are more flexible and able to adopt more 3D conformations [43] (Figure 4).

The systematic evolution of ligands by exponential enrichment (SELEX) is used to isolate aptamers with a high affinity for a molecular target from approximately 10^12^–10^15^ oligonucleotide combinations [45,46,47]. In general, the SELEX process consists of three steps: creating a “library”, binding and separation, and amplification. This process is repeated to find the nucleotide with the best specificity and affinity.

### 3.4. Comparison of Aptamer with Antibodies

Aptamers are often called synthetic antibodies and can mimic antibodies in a number of applications. The selected aptamer binds to a specific target with a comparable affinity and specificity to that of an antibody [47]. Aptamers present several advantages compared to antibodies as bioreceptors, namely the production of aptamer that is relatively easy and economical with conventional chemical methods compared to antibodies, which are more expensive because they usually require the use of animals as hosts during the production process [48]. Aptamers can interact very strongly with the desired target because conformational changes can occur during the binding process when using various bonds [49]. Aptamers can bind to target compounds via noncovalent multipoint interactions based on the combined contribution of several attractive forces, including hydrogen bonding, electrostatic and hydrophobic interactions, π–π arrangement, and Van der Waals forces. This multi-point interaction is favored by a conformational change in the secondary structure of the aptamer, which folds back around the target with complementary molecular shapes and chemical structures. This binding mechanism allows for the formation of highly stable affinity–target aptamer complexes, with dissociation constants ranging from the nanomolar level to the picomolar level [50] (Table 2).

An important step in the development of the electrochemical aptasensor is the mobilization of the aptamer to the electrode surface and the development of a strategy for reliable aptamer mobilization so as to maintain its biophysical characteristics and binding ability, as well as to minimize nonspecific binding/adsorption events [52]. As the sensor requires the aptamer to be mobilized to a solid surface, many modifications have been made; for example, the addition of a linker in the form of a thiol group. The role of the linker is to spatially extend the receptor from the surface to increase its accessibility by ligand solutions and eliminate non-specific adsorption. The type of coadsorbent affects the reduction in non-specific binding and maintains the integrity of the being mobilized biomolecules [53].

## 4. Aptasensor for Diagnosis of Diabetes Mellitus and Mitochondrial Diabetes

### 4.1. Aptasensor for Glucose Detection

Blood glucose is a carbohydrate monomer that is absorbed by the body and typically stored as glycogen in the liver and skeletal muscles. In the body, glucose serves as a source of energy and is metabolized, with one example being glycolysis. In patients with DM, hyperglycemia occurs due to insulin dysfunction or insulin resistance, preventing glucose from entering the cells [54].

Liu et al. (2022) developed a portable chip aptasensor with smartphone measurement for the precise diagnosis of prediabetes/diabetes by monitoring glucose and insulin. In their study, thiol group-bound aptamers were immobilized through Au–S bonds to gold nanoparticles on screen-printed carbon electrodes (SPCEs), which then bound to the methylene blue redox probe. The detection was carried out using saliva samples, and the biochip was able to detect glucose concentrations as low as 0.08 mM. The results indicate that the detection biochip performs well when detecting the desired glucose levels [55].

### 4.2. Aptasensor for HbA1c Detection

HbA1c is hemoglobin that binds glucose at the N-terminus of the amino valine of the hemoglobin β chain. This structural change results in a change in the secondary structure of Hb so that it has a different polarity from Hb [56] (Figure 5).

Glycated hemoglobin is formed through a non-enzymatic reaction. The first stage of the formation of HbA1c is the interaction of hemoglobin with the aldehyde group of glucose to form an aldimine covalent bond called the Schiff base and the second stage is irreversible where the Schiff base rearrangement occurs to become a more stable ketoamine and is called the amadori product [58]. HbA1c was recommended by the American Diabetes Association (ADA) in 2010 as a biomarker for diabetes diagnosis and blood sugar control because it is considered capable of identifying average plasma glucose concentrations over the past 60–90 days to reduce complications [59]. HbA1c is also a very potential biomarker because its blood level is not affected by the time of day, recently eaten food, fasting, and rapid lifestyle changes [60]. The level of formation of this glycated protein can be used to assess glycemic control and diagnose type 2 diabetes [61]. The higher the HbA1c level, the more glucose binds to hemoglobin [62].

The detection technique that is currently being developed uses an electrochemical aptasensor for HbA1c detection because of its advantages, such as having a high affinity, low cost, easy modification and specific response [63,64]. Eissa et al. (2017) developed a label-free aptamer array to detect HbA1c and tHb in human blood. This aptamer is specifically designed to interact with HbA1c; the aptamer is modified by adding a thiol group at the end and will be immobilized on the surface of the AuNP electrode, which can then be used as a method of detecting HbA1c and tHb by voltammetry. This aptasensor showed high sensitivity, with detection limits of 0.28 and 0.33 mg/mL, respectively [65].

Eissa et al., in 2019, showed high selectivity and sensitivity results for HbA1c. The existence of a specific binding between the target protein (HbA1c) to the aptamer can increase the transfer of electrons on the surface of the electrode and produce a certain signal. This is the basis for measurement. The resulting signal is directly proportional to the concentration of HbA1c in the sample [66].

Research was carried out by Anand et al. (2021) regarding docking and molecular dynamics’ simulations to understand the high-affinity interaction between the aptamer and the target. In his research, the 3D structure of the SELEX aptamer results was modeled, which was then docked to determine the binding site to the peptides aglycated with the four fructose D tautomers as ligands (indicated by G1, G2, G3, and G4). The results of the docking showed three interaction sites on the aptamer, namely hairpin 1, hairpin 2, and loop. Compared to the three binding sites, hairpin 1 has a higher docking score than the other binding sites, of −9.2 kcal/mol [9] (Figure 6).

To further understand the binding interactions, molecular dynamics simulations were conducted for 70 ns between the aptamer and GP1, GP2, GP3, and GP4. The resulting complex showed a structural stability with an RMSD value of approximately 1.0 A. Hydrogen bonds were observed at GP1 between the hydroxyl groups of fructose and peptides, such as histidine2, thirosin4, and glutamine6, with the aptamer nucleotides, namely G2, G4, A5, A17, G18, and C33. These interactions are illustrated in Figure 3. The free energy calculated from the simulation results indicated that GP1 had the highest energy value of −12.50, while GP2 had the lowest free energy value of −10.79. The MD simulation free energy was then compared with the experimental results obtained by SiNW-FET, which showed a free energy value of 12.30 ± 0.05 kcal/mol for the aptamer complex and GP.

### 4.3. Aptasensor for GHSA Detection

GHSA is an albumin that is covalently bonded to a glucose group. The formation of this glycate is non-enzymatic, where the amino acids lysine, arginine, and cysteine in albumin covalently bind to the aldehyde group of sugar to form an irreversible Schiff base so that it undergoes Amadori rearrangement to form a more stable aminomethyl ketone (ketoamine). Under normal circumstances, glycated albumin levels range from 6 to 15%, but in DM patients it increases by about 2–3 times [67].

GHSA can be used as medium-term glycemic control by reflecting glycemic status in the previous 2–3 weeks and is not affected by hemoglobin metabolism and erythrocyte age. GHSA was chosen as a faster control of glycemic status than HbA1c, especially for patients suffering from anemia, patients receiving iron preparations and pregnant women (associated with iron-deficiency anemia) [68].

Increased glycated albumin levels are associated with the development of vascular complications in diabetic patients, particularly those with chronic kidney failure who undergo hemodialysis. Glycated albumin can form advanced glycation end-products (AGEs), which can produce reactive oxygen species (ROS). These ROS can bind to cell surface receptors and form cross-links, leading to micro- and macro-vascular complications in diabetes. AGEs can also alter the function of intracellular proteins, change the extracellular matrix, and affect hormone action, cytokines, and free radicals through cell-surface receptors [69]. In a study conducted by Pu et al. in China with 320 subjects, elevated GHSA levels were associated with the presence and severity of coronary artery disease [70].

Albumin examination can be used to assess short-term glycemic control in certain conditions where HbA1c measurement cannot be applied, such as in patients with type 1 diabetes, type 2 diabetes receiving insulin therapy, hemolytic anemia, bleeding, blood transfusion, chronic kidney failure undergoing hemodialysis, pregnant women, liver cirrhosis, variant hemoglobin, and patients with postprandial hyperglycemia [68]. However, this examination cannot be used for patients affected by albumin metabolism disorders such as those with nephrotic syndrome, hyperthyroidism, and those receiving glucocorticoid treatment [68].

Research has been conducted to measure GHSA levels using interdigitation electrode-based disposable enzyme sensor strips that show excellent sensitivity with high reproducibility [71]. In addition to the use of enzymes as bioreceptors, many biosensors have been developed for GHSA detection using aptamers because of their advantages. Gosh et al. (2017) described the diagnosis of diabetes mellitus using aptasensor-based optical detection. This aptasensor uses ssDNA aptamers, quantum dot semiconductors and gold nanoparticles. The detection principle will result in an increase in photoluminescense in line with the increase in GHSA concentration. This diagnosis is effective if made in conjunction with traditional methods of glucose monitoring [72]. In addition to using gold nanoparticles, it is also possible to use phosphorescent quenching graphene oxide (GO) for the GHSA detection aptasensor. The ability of graphene oxide to interact with single-stranded DNA (ssDNA) molecules was proven through the increase in π–π and the ability to quench the fluorescence intensity of ssDNA-labeled fluorescence. Go and fluorescently labeled aptamers are incubated to form a GO–aptamer complex (showing off signal). When GHSA is added to the reaction, the fluorescent label of the aptamer is released from the GO surface and binds to GHSA, producing a fluorescence signal [73,74].

The detection method developed by Bunyarataphan in 2019 uses an aptamer-based electrochemical biosensor, which interacts specifically with GHSA and HSA and measures protein-binding to the aptamer using square wave voltammetry (SWV). The aptamer is modified by adding biotin and immobilized with SPCE, which is then added with streptavidin. During the measurement process, it is expected that the aptamer will bind to GHSA or HSA, forming a complex that inhibits the electron transfer rate, which is then measured as an electrochemical signal. The aptamer specificity test was conducted by reacting the aptamer with other interfering substances in the blood, namely GHSA, HSA, glucose, glycine, folic acid, and ampicillin [75].

In another study by Aye et al. (2021), the electrochemical aptasensor was studied, focusing on the sensitive and selective detection of albumin glycation in DM patients with thalassemia. The specificity tests that were conducted were carried out with other biomolecules that cause reduced biosensor performance, such as glucose, HAS, and bilirubin. There are also drugs that are used as comparators, such as ampicillin and folic acid. Experimental results show a high specificity in the detection of GHSA [76]. Zhou et al. (2023) conducted research on a flexible multielectrode array-based electrochemical aptasensor for the simultaneous detection of HSA and GHSA. This aptasensor, with HSA and GHSA targets, has the lowest detection limit of 13 nm for HSA and 25 nm for GHSA. This target shows good selectivity in diluted whole blood samples. This aptasensor utilizes a polymer substrate for the sensor chip that can reduce material costs, thus providing a reliable and affordable long-term glycemic control tool at the point of care [77].

### 4.4. Aptasensor for Insulin Detection

A hormone that plays an important role in glucose metabolism is insulin. Insulin has two chains, namely the A chain with 21 amino acids and the B chain with 30 amino acids with a molecular weight of 5802 and an isoelectric point of pH5.5. Both are connected by two disulfide bonds from the N-helix in the A chain to the center of β and the C-terminal from the A chain to the center of the B chain [78].

Insulin synthesis occurs in the beta cells of the pancreas. Initially, insulin is in the form of a signal peptide, which is synthesized into pre-proinsulin and then becomes proinsulin in the endoplasmic reticulum. Secretory vesicles will send proinsulin from the endoplasmic reticulum to the golgi apparatus, and zinc and calcine are added so that proinsulin forms a hexamer that is not soluble in water. This proinsulin hexamer is converted into insulin and C-peptide with the help of enzymes outside the golgi apparatus [78].

Yoshida et al. (2009) conducted a selection of aptamer arrays that can detect insulin using the SELEX method, resulting in aptamers with higher affinity compared to insulin-link polymorphic repeat region (ILPR) [79]. The arrangement of aptamers chosen by Yoshida was used by research by Kubo in 2015 to describe the electrochemical aptasensor immobilized on a gold electrode to determine its electrochemical activity. The results showed that the peak current depends on insulin concentration [80].

A detection method was developed using the aptasensor by Zhao et al. (2019) regarding the dual-signaling aptasensor for the ultra-sensitive detection of insulin in order to obtain a sensor with a high level of sensitivity and specificity. In the biosensor device, an insulin binding aptamer modified with methylene blue is used as a signal-off probe and AuNP modified with DNA (Ferrocene) as a signal-on probe, both of which are integrated with the mDNA linker to make the AU electrode a sensing interface. The MB and Fc responses are used as signal indicators. This sensor scheme is expected to have a detection limit for insulin with a low concentration and high specificity [81] (Figure 7).

A specificity test was carried out with thrombin, HSA, and HigG because it was considered a substance that interfered with the target during the detection process; the results showed that the proposed aptasensor specificity was very good at detecting targets. This aptasensor has an acceptable detection stability and exhibits comparable aptasensor reproducibility with a single signaling.

In 2021, Asadpour et al. (2021) conducted research monitoring the aptasensor to detect insulin using thin films of mesoporous silica nanoparticles using the EASA method. This aptasensor has a detection limit of 10–350 nM and shows good selectivity for glucose, urea, glutathione, and dopamine [82].

### 4.5. Aptasensor for ATP Detection

ATP, which is called the energy currency in bioenergetics, can be used to evaluate or detect cell viability and proliferation [83], cell death [84,85], and energy transmission [86]. Manson et al. (2012) and Ellsworth et al. (2009) explained that ATP is used as a compound that indicates a breakdown in metabolism [87,88]. The three main pathways that produce ATP in eukaryotic organisms are glycolysis, the citric acid cycle/oxidative phosphorylation, and the beta-oxidation of fatty acids. When the production of the process of ATP synthesis in mitochondria is disrupted, it causes disease, one of which is mitochondrial diabetes [89].

ATP is a biomarker for mitochondrial diabetes. Naing et al., in 2014, suggested that mitochondrial diabetes could be caused by a mutation in the mitochondrial organelle DNA component at position 3243 [90]. Maksum et al. (2013) described the pathology of the A3243G mutation, which causes the inhibition of Leusil-tRNA synthetase activity, which can form early respiration proteins that can inhibit ATP production [91]. When this process is inhibited, mitochondria’s important role in the production of energy in the form of ATP will be disrupted, resulting in cellular abnormalities and cell death [92]. Additionally, the A3243G mutation can affect the methylation process at the G10 nucleotide position, where the A nucleotide at position 3243 is responsible for introducing methyltransferases. Failure of this process can lead to a lack of methylation, which is critical because a single missing methyl group can affect the binding of tRNA to other components that play a crucial role in the machinery of mitochondrial protein synthesis.

A molecular dynamics simulation study was conducted on the secondary structure prediction approach of the tRNA-leucine dimer of the A3243G mutation. The results predicted stability in terms of binding energy and root mean square deviation (RMSD). The results show that the RMSD value of the dimer structure of the mutant leucine tRNA is lower because there are more hydrogen bonds than there are on a normal dimer [93]. Recent research has proved in silico the effect of the mitochondrial DNA tRNA leucine A3243G mutation in patients with type 2 DM. The study showed that the stability of the dimer structure of the mutant is due to the presence of nucleotide bonds 14 and 48. The stable dimer structure resulted in a decrease in the process of aminosylation [94].

Furthermore, the T10609C and C10676G mutations in the ND4L gene are associated with proton translocation in patients with type 2 diabetes and cataracts. Molecular dynamics simulations were performed on the native and mutant subunits, and the simulation results showed that the mutations resulted in the disruption of the translocation pathway by the formation of hydrogen bonds between Glu34 and Try157 [95]. Several mutation detection tests have been developed, including a study that designs and optimizes PCR-RFLP tests for the detection of G9053A and T15663C mutations, as well as other studies on the confirmation of point mutations using site-directed mutagenesis for mutations in mitochondria [96,97]

In addition to the mutations in A3243G, Maksum et al. (2017) also explained the mutations in G9053A that are found in diabetes and cataract patients. The result of these mutations can interfere with the ATPase6 proton translocation channel. The mutation changes electrostatic potential from serine to asparagine, causing the inhibition of insulin secretion. This results in hyperglycemia as the glucose in blood cannot be taken up by GLUT and remains in the blood [34,98]. ATP can be considered as a suitable biomarker for mitochondrial diabetes.

The ATP molecule is composed of three subunits: adenine base, ribose sugar, and triphosphate. The triphosphate is the reactive site due to the low activation energy required to break the phosphodiester bond, which is relatively weak. This allows for the final phosphate group to be easily transferred from one molecule to another while simultaneously releasing energy [99]. However, the selectivity of aptamers for ATP is not very good for other adenosine phosphates, such as ADP and AMP. Nutiu and Li (2003) found that the ABA aptamer shows a relatively high affinity for adenine, but not the triphosphate group, meaning that adenosine, AMP, and ADP provide the same fluorescence yield response as ATP.

Efforts have been made to optimize the design of aptamer structures so that they can interact specifically with nucleobases and triphosphates. However, some unfavorable results have been reported. Sazani et al. (2014) found that, although the aptamer has a good level of interaction with triphosphate, its affinity for nucleobases is reduced [100]. Another attempt to increase selectivity takes advantage of the difference in density between ATP, ADP, AMP, and adenosine by introducing a secondary recognition mechanism that can strongly interact with the triphosphate moiety of ATP, such as the uranyl salofen complex [101,102,103]. To prevent the repulsive interaction of the triphosphate group with the anionic back of the aptamer, the use of neutral nucleic acids such as peptide nucleic acids or morpholinos can be modified.

One of the efforts made by Kheyrabadi and Mehrgardi (2013) in the application of aptamer for electrochemical biosensors is to fragment ABA into two aptamer fragments with a termination point between T14 and T15 [104]. The aptamer that can interact with ATP is the ATP-binding aptamer (ABA) with PDB ID: 1AW4. The structure of the aptamer shows that there are two recognition pockets, one consisting of G6, G22, and A23, and the other containing G9, A10, and G19. Each pocket can hold one ATP molecule [105].

The aptamer was fragmented into two parts: F1 with a 5′ end plus an alkyl and a thiol conjugated with AuNP, which was used to increase the sensitivity and conductivity of the analyte signal due to its large surface area and excellent electrocatalytic ability [106]. The covalent interaction strength of the thiol–gold coordination provides the basis for fabricating strong, self-assembled monolayers, which can be used for a wide range of applications and thus have enabled many important achievements in nanobiotechnology [107]. This research also measures the sensor signal with other ATP analogues, resulting in a specific response to ATP, and has acceptable stability.

From the results of the selectivity test, compared to UTP, GTP, and CTP with the same concentration, F1 and F2 aptamers showed good selectivity, with an ATP 10 signal that was times higher than the other analogues. Recent research from Mulyani in 2022 also supports the research of Kashefi-Khernyabadi (2013); Mulyani conducted electrochemical sensor research using the ATP-binding aptamer with the differential pulse voltammogram (DPV) method. The voltammogram characterization results show that the peak current in ATP is lower than in UTP, CTP, and GTP [108].

Rustaman et al. (2023) conducted research on the specificity of aptasensors for diabetes mellitus detection using in silico methods. This in silico research refers to the research of Kashefi (2013) to determine the interaction of aptasensor candidates with ATP through the interaction site and interaction stability, and also determine the specificity of the aptamer by comparing its interactions with ADP and AMP.

This study analyzed the interaction based on a molecular dynamics simulation for 100 ns. The significant interactions that occurred were three hydrogen bonds between ATP and G7, G8, and A24. In particular, the ATP aptamer complex has a quite good interaction and better specificity potential than ADP and AMP. Therefore, this study shows that the aptamer for ATP detection is very promising for further tests, such as tests on direct samples and clinical trials of the aptasensor as a diagnostic tool [109].

Research was carried out by Xie et al. (2020) regarding in silico ATP-binding aptamer (ABA) studies. Simulations revealed that the aptamer displays a high degree of rigidity and is structurally very little affected by ATP binding. The decomposition of interaction energies suggests that the dispersion forces from deposition between ATP and nucleobases G6 and A23 at the aptamer binding sites play an important role in stabilizing the supramolecular complex, as do the hydrogen bond interactions between ATP and G22. In addition, metadynamic simulations show that, during the association process, water molecules act as an important bridge connecting ATP with G22, which supports the dynamic stability of the complex [105].

An explanation of the various apatamers used for diabetes mellitus biomarker detection is summarized in the following Table 3.

## 5. Conclusions and Future Direction

This review summarizes the recent development of aptasensors for the diagnosis of biomarkers of diabetes mellitus, namely glucose, insulin, HbA1c, GHSA, and ATP. These aptasensors can be specifically designed to bind to specific biomarkers to distinguish between types of diabetes mellitus (type 1, type 2, and mitochondrial diabetes). An research objective in the field of biosensors is to produce an aptasensor for biomarker detection for use by the community as an early diagnostic tool for diabetes mellitus that can distinguish between types of diabetes mellitus so that there are no treatment errors that result in death.

The future direction of the aptasensor faces some challenges related to the reliability and validity values of the diagnostic tool, such as sensitivity, stability, and specificity. In addition, aptamer interaction studies should be conducted with the target and pharmacokinetics before clinical trials and commercialization.

## Figures and Tables

**Figure 1 diagnostics-13-02035-f001:**
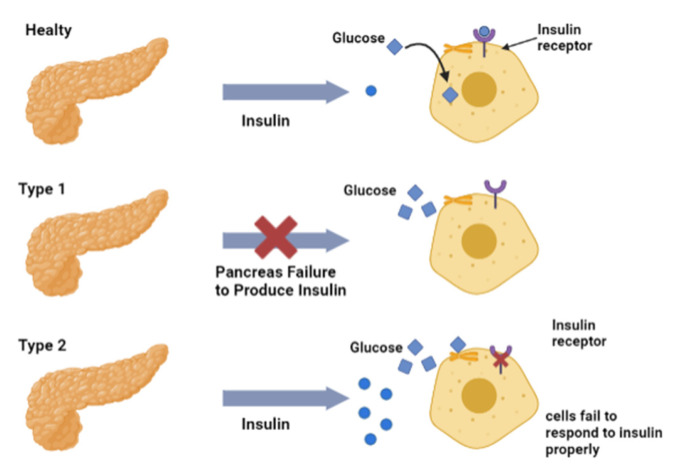
The types of diabetes: type 1 diabetes due to insulin secretion and type 2 diabetes due to insulin resistance.

**Figure 2 diagnostics-13-02035-f002:**
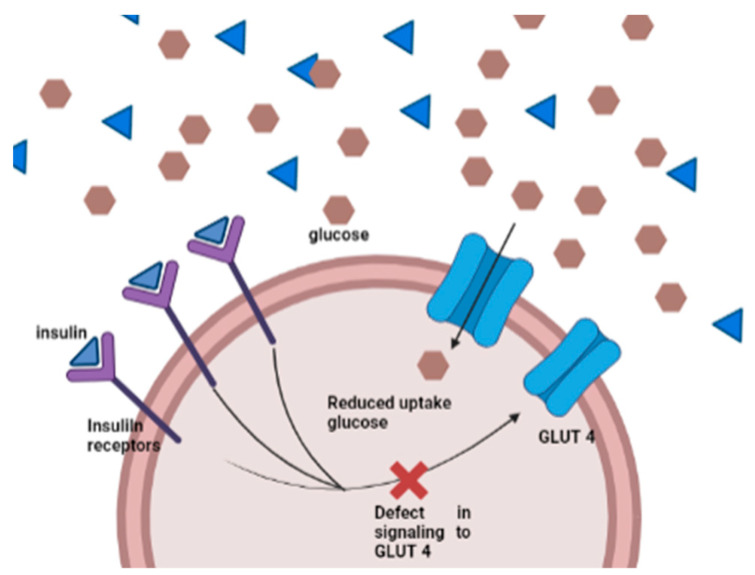
Pathology of type 2 diabetes caused by insulin resistance [33].

**Figure 3 diagnostics-13-02035-f003:**
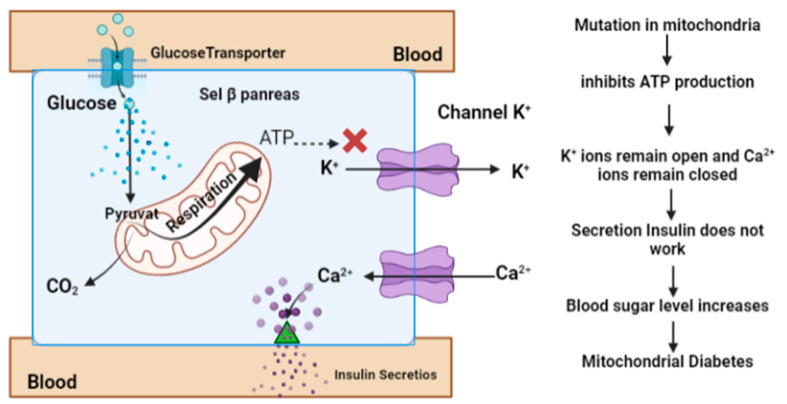
Effect of the regulation of ATP concentration levels on the insulin secretion and pathology of mitochondrial diabetes [35].

**Figure 4 diagnostics-13-02035-f004:**
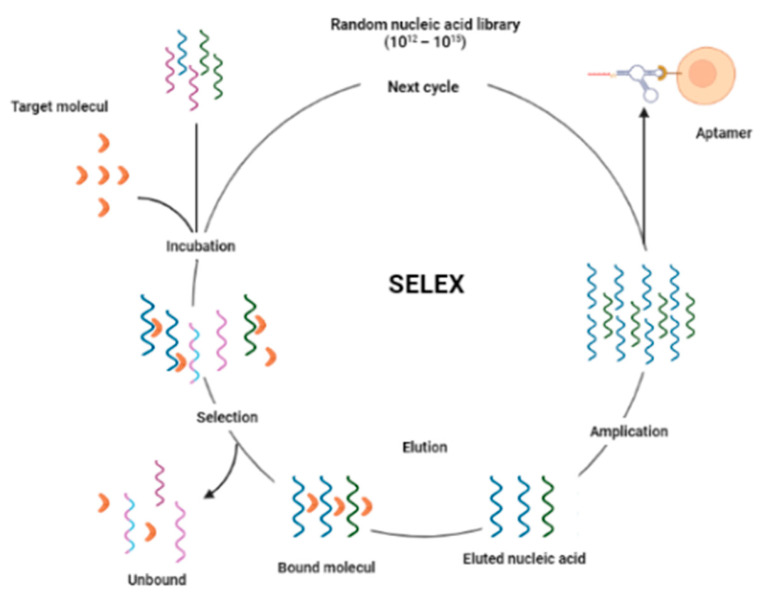
SELEX scheme. The process begins with incubation of the target with a random nucleic acid library, followed by selection and elution to obtain a specific aptamer against the target, and then amplificatin by PCR to obtain the aptamer library [44].

**Figure 5 diagnostics-13-02035-f005:**
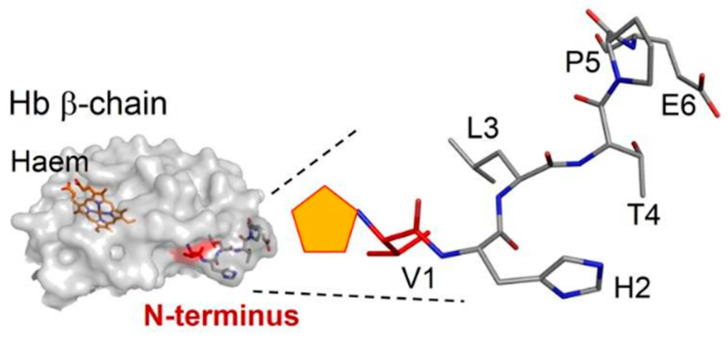
Left, 3D structure of hemoglobin β-chain, the prosthetic group of hemoglobin is marked in orange. Right, magnification of the structure of the amino acid glucose-binding site at the N-terminal of the Valine β-chain (in red). The orange pentagon at the N-terminal valine amino group indicates a fructosyl group [57].

**Figure 6 diagnostics-13-02035-f006:**
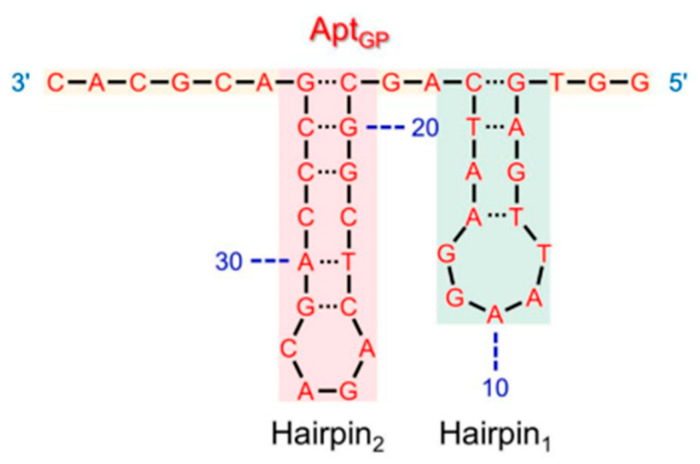
The secondary structure of the aptamer selected by the SELEX method shows the hairpin 1 (blue colored) and hairpin 2 (bright red colored) regions with the highest binding affinity to the target [9].

**Figure 7 diagnostics-13-02035-f007:**
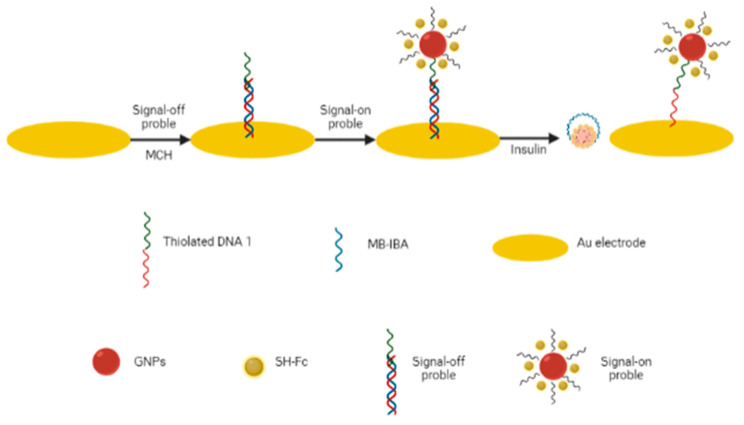
Schematic representation of insulin detection using an electrochemical dual-signaling aptasensor [81].

**Table 1 diagnostics-13-02035-t001:** Criteria for diagnosis of diabetes mellitus.

	Current Plasma Glucose (mg/dL)	Glucose Tolerance Impaired (mg/dL)	Fasting Blood Glucose (mg/dL)	HbA1c (%)
Diabetes	≥200	≥200	≥126	≥6.5
Prediabetes	Not applicable	140–199	100–125	5.7–6.4
Diabetes during pragnency	≥200	≥200	≥126	≥6.5
Normal	Not applicable	<140	<100	<5.7

**Table 2 diagnostics-13-02035-t002:** Comparison of aptamers with antibodies [51].

Characteristics	Aptamer	Antibody
Toxic or poorly immunogenic immunogenogenic target	No	Yes
Isolation process	In vitro selection under various conditions	Limited to psychological conditions with animal immunization
Immunogenicity	Little to none	Significant
Batch activity	Uniform	Varies
Stability	Insensitive to redox reactions. Difficult to aggregate due to lack of hydrophobicity. Resistant to changes in pH and temperature.	Sensitive to redox reactions. Easy to form aggregates. Sensitive to changes in pH and temperature.
Screening time snd cost	Limited screening processes	Time-comsuming and expensive
Shelf life	Long	Limited
Controllable binding and release condition	Can be changed on demand	Difficult to modify
Chemical modifications	Easy to modify with low cost	Difficult and expensive to modify

**Table 3 diagnostics-13-02035-t003:** Summary of aptasensors used to detect diabetes mellitus biomarkers.

Detection Principle	Biomarker	Aptamer Sequence	Result	Ref
Surface modification of SPCE with gold nanoparticles using methylene blue as a redox probe. The prepared SPCE is integrated with a micro-device containing a bluetooth transmission system for smartphone signal reading.	Glucose and Insulin	Glucose: 5′–HS–HS-C_6_-CTCTCGGGACGACCGTGTGTGTTGCTCTGTAACAGTGTCCATTGTCGTCCC-MB-3′ Insulin: 5′ –HS–HS-C_6_-AAAAGGTGGTGGGGGGGGTTGGTAGGGTGTCTTCT-MB3′	Good stability and high sensitivity.	[55]
Apatamers modified with thiol groups were immobilized on AuNP electrodes and measured using square wave voltammetry (SWV).	HbA1c	5′-GGGGACACAGCAAC ACACCCACCCACCAGCCCCAGCATCATGCCCATCCGTCGTGTGTG-3′	High sensitivity and selectivity, with an LoD of 0.2 mL^−1^ and a linear range of 100 pg mL^−1^–10µg mL^−1^.	[65]
The aptamer was non-covalently immobilized on six electrodes of nanomaterials via π–π stacking interactions between DNA nucleobase and carbon material surface. Measurements were performed using SWV.	HbA1c	5′-ACACACCCACCCA CAGCCCCAGCATCATGCCCATCCGTCGTGTGT-3′	The SWCNT aptasensor showed the highest selectivity and sensitivity, with a limit of detection (LoD) of 0.03 pg mL^−1^.	[66]
Aptamer immobilization on silicon nanowire field effect transistors. Aptamer was also modified with polyethylene glycol. Docking and molecular dynamics simulations were performed.	HbA1c	5′-HS-GGT GAG TTA AGG AAT CAG CGG CTC AGA CGA CCC GAC GCA C-3′	Aptamer has a binding energy of 12.30 ± 0.05 kcal/mol, which is consistent with theoretical calculations.	[9]
Immobilization of thiolated aptamer on gold electrode surface and quantum dot. Measurement conducted using Ocean Optics USB4000 spectrophotometer.	GHSA	5′-Thiol C6/TGCGGTTGTAGTACTCGTGCCCG/Thiol C_6_ SS 3’	High selectivity to GHSA. Concentration range of 1008 nM–4500 nM and limit of detection of 1 nM.	[72]
Immobilization of fluorescently labeled aptamer on graphene oxide surface. When GHSA is present, the fluorescence-labeled aptamer will bind to GHSA and produce a signal.	GHSA	5′-GGTGCGGTTCGTGCGGTTGTAGTACTCGTGGCCGATAGAGGTAGTTTCG-3′	High sensitivity when detecting human serum with an LoD of 50 μg mL^−1^	[73]
Immobilization of biotinylated aptamer on screen-printed carbon electrode modified with streptavidin and measurement using SWV.	GHSA	5’ TGCGGTTGTAGTAC TCGTGCCCG-3	High selectivity for GHSA compared to glucose, glycine, folic acid and ampicillin.	[75]
Immobilization of thiol aptamer on the surface with graphene oxide was measured using SWV.	GHSA	5′ -NH_2_-TGC GGT TGT AGT ACT CGT GGC CG–3	High selectivity, with an LoD of 0.031 μg mL^−1.^	[76]
Immobilization of aptamer on the surface of the pollimer chip modified with gold electrode. Measurement conducted by differential pulse voltammetry (DPV).	GHSA and HSA	GHSA: 5′-ferrosen-(CH_2_)6-GTC TCA GCT ACC TTA CCG TAT GTG GCC CAA AGC GTC TGG ATG GCT ATG AA-(CH_2_)6-SS-(CH_2_)6-OH-3′ HSA: 5′-ferrosen-(CH2)6-TGC GGT TCG TGC GGT TGT AGT ACT CGT GGC CGA T-(CH_2_)6-SS-(CH_2_)6-OH-3′.	Selective to the target in diluted blood samples with an LoD for HSA of 13 nM and 25 nM for GHSA.	[77]
Selection of aptamers by SELEX process; the process of measuring circular dichroism (CD) is used to select specific aptamers.	Insulin	5′-GGTGGTGGGGGGGG TTGGTAGGGTGTCTTC-3′	The aptamer folded well into a quartet structure.	[79]
The aptamer was immobilized on a gold electrode. Measurements were made using cyclic voltammetry and activity was confirmed using spectrophotometry.	Insulin	5′-GGTGGTGGGGGGGGTTGGTAGGGTGTCTTC-3′	The cathodic peak in IGA3–hemin complex decreased with insulin concentration.	[80]
Immobilization of insulin aptamer on graphene oxide with quaternary tetraphenylethine salt probe.	Insulin	5′-GGTGGTGGGGGGG GTTGGTAGGGTGTCTTC-3′	Good linear relationship, with an insulin concentration ranging from 1.0 pM to 1.0 μM with a low LOD of 0.42 pM.	[110]
Immobilized insulin binding aptamer (IBA) modified with methylene blue (MB) as “signal off” probe and aptamer/ferrocene modified with gold nanoparticles as “signal on” probe. Measurements were made using SWV.	Insulin	Aptamer IBA: 5′-SH-TTTTTTCAC CCT ACC ACC CCCTATGTAATA AGA GCT AAA-3′	High sensitivity and selectivity; good reproducibility with an LoD of 0.1 pM and linear range of 10 pM–10 nM.	[81]
The aptamer hybridizes with cDNA on the mesoporous silica surface as a molecule that limits the diffusion of the probe towards the electrode by closing the mesochannels. When binding to insulin, the hybridization of cDNA with the aptamer will be destroyed and open the nano-channels, resulting in an increase in DPV signal.	Insulin	5′-GGT GGT GGG GGG GGT TGG TAG GGT GTC TTC-3′	High selectivity and sensitivity, with an LoD of 3 nM and linear range of 10.0–350.0 nM.	[82]
Immobilization of aptamer-labeled red emission carbon dots (R-CDs) on graphene oxidase. Effectively detected insulin (INS) generates fluorescence resonance energy transfer (FRET).	Insulin	5′-(CH_2_)6-NH2GGT GGT GGG GGG GGT TGG TAG GGT GTC TTC-3′	High sensitivity and anti-interference, with an LOD of 1.1 nM and a linear range of 1.3–150 nM.	[111]
Immobilization of aptamer on glass carbon electrode (GCE); when insulin binds to aptamer, it will undergo hydrolysis with the help of Exo 1 to cut the aptame that does not bind. Gold nanoparticle-probe aptamer is added so that a complex is formed to form a sandwich assay. Measurement with DPV.	Insulin	5′-SH-(CH_2_)6-GGT GGT GGG GGG GGT TGG TAG GGT GTC TTC 3′	High selectivity and sensitivity for the detection of insulin, with an LoD of 9.8 fM and linear range from 0.1 pM to 1.0 μM.	[112]
Immobilization of aptamer on quantum dot surface. Fluorescence signal measurement.	Insulin	NH_2_-5’-GGT GGT GGG GGG GGT TGG TAG GGT GTC TTC-3’	High sensitivity and selectivity, with a linear range of 0.001–5000 nM and LoD of 0.5 pM.	[113]
Methylene blue (MB)-modifed insulin-specifc aptamer immobilized on gold-deposited screen-printed electrodes.	Insulin	SH-(CH2)6–GGT GGT GGG GGG GGT TGG TAG GGT GTC TTC—AttoMB2	The stability of the sensor was good for 10 days, remaining at 92% with high sensitivity and selectivity.	[114]
Aptamer with MB probe immobilized on Au@Fe-MIL-88. Measurements by cyclic voltametry and DPV.	Insulin	5′-NH2 GGTGGTGGGG GGGGTTGGTAGGGTGTTTTC-3′	High-sensitivity dan reproducibility, with an LoD 1.3 × 10^−16^ mol L^−1^.	[115]
Immobilization of aptamer on modified SPCE on gold nanoparticles. Measurement with DPV.	ATP	F1: 5′-HS-(CH_2_)6-ACCTGGGGGAGTAT-3′ F2: 5′-TGCGGAGGAAGGT-CH_2_)2-NH_2_-3′	High selectivity for ATP compared to UTP, CTP, and GTP. Good reproducibility and stability.	[104]
Immobilization of aptamer on modified SPCE on gold nanoparticles. Measurement with DPV.	ATP	F1: 5′-HS-(CH_2_)6-ACCTGGGGGAGTAT-3′ F2: 5′-TGCGGAGGAAGGT-CH_2_)2-NH_2_-3′	High selectivity for ATP compared to UTP, CTP, and GTP, with LOD and LOQ values of 7.43 and 24.78 μM, respectively.	[108]
Molecular dynamics simulation between aptamer and ATP, forming a sandwich conformation assay to determine the interaction.	ATP	F1: 5′-HS-(CH2)6-ACCTGGGGGAGTAT-3′ F2: 5′-TGCGGAGGAAGGT-CH_2_)2-NH_2_-3′	High specificity to ATP compared to ADP and AMP.	[109]
Molecular dynamics simulation between aptamer and ATP to determine the interaction and binding energy.	ATP	5′-ACCTGGGGGAGTAT TGCGGAGGAAGGT-3′	Aptamer has a high degree of rigidity due to the influence of ATP. Interacts on nucleobases at G6, A23, and forms hydrographic bonds at G22.	[105]
The aptamer was immobilized on a modified glassy carbon electrode. Measurement conducted by DPV.	ATP	5′-NH_2_-ACC TGG GGG AGT ATT GCG GAG GAA GGT-3′	High selectivity, stability and reproducibility with an LoD of 0.01 pM and linear range of 0.01 pM–1 μM.	[116]
Immobilization of aptamer on modified GCE with Fe3HAI4@ Cu@Cu2O nanocomposite.	ATP	5′-NH_2_-TGGAAGGAGGCGTTATGAGGGGGTCCA-3′	High sensitivity and excellent specificity, with an LoD of 0.17 nmol/L and a linear range of 0.5–2500 nmol/L.	[117]
Immobilization of aptamer labeled with FAM aptamer on origami-based duplex.	ATP	5′-CACTGACCTGGGGGAGTATTGCGGAGGAAGGT-3′	High sensitivity and selectivity with LoD 0.29 ng mL^−1^ and a linear range of 0.1 ng mL^−1^.	[118]

## Data Availability

Not applicable.

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
