# Peer review of "Detection of Biomarker Using Aptasensors to Determine the Type of Diabetes"

_diagnostics, 2023, doi:10.3390/diagnostics13122035_

Round 1

Reviewer 1 Report

There are always reviews on the topic, e.g.: „A Critical Systematic Review of Developing Aptasensors for Diagnosis and Detection of Diabetes Biomarkers“ (Crit Rev Anal Chem. 2022;52(8):1795-1817. doi: 10.1080/10408347.2021.1919986). 

Some missing references are e.g.:

·         doi: 10.1007/s10404-022-02622-3

·         doi: 10.1016/j.snb.2023.133730

·         doi: 10.1038/s41598-019-57396-6

·         doi: 10.1080/10408347.2021.1919986

·         doi: 10.1088/1361-6528/aa893a

·         doi: 10.1109/NANO.2015.7388933

·         doi: 10.1016/j.bios.2016.04.015

·         doi: 10.3390/molecules26030734

The introduction on general aspects of aptasensor and aptamer is too long. The authors should spend more effort on actual papers from 2021 and 2023, only 12 of them are present in the actual manuscript.

Reviewer 2 Report

‘’ Detection of Biomarker Using Aptasensors To Determine The 2 Type of Diabetes’’

The manuscript presented for review consists of 17 pages with 100 references. 2 tables and 4 figures are included. The manuscript is divided into 2 sections (Introduction, Discussion). What were the methods of the study?What are the conclusions, steps for the future? The work fits the journal scope. The subject is relevant for the field. English language is understandable - minor editing is required (for example a repetition of "due to" - line 30, a word "namely", style, grammar structures). Keywords are adequate and refer to the whole context. I would suggest to add keywords: detection methods/ biosensors. The aim of the study is defined in line 16-19. The references are adequate, most of them are recent. Are there any studies from 2023 concerning the subject? 

An abstract:

1)Line 10 - "this type of diabetes" - which type? 

2)I would recommend to explain abbreviations "ATP","GHSA","HbA1c" in brackets. There is no need to repeat explanation in the whole text of the study. 

Introduction and Discussion:

1)It could be helpful to add a figure which shows clearly the types of diabetes. 

2)"Blood glucose" - I would included OTGG test, fasting glucose level...

3)Line 53-80 - Most of the text is similar to information described in Introduction. 

4)Table 1 - in Line 58 the Authors mentioned about diabetes during pregnancy. Could you include criteria in the table? "N/A" - what does this symbol mean? There is no explanation below the table, in the legend.

5)Could you create a figure with pathogenesis of the disease? It would be very helpful to present the phenomena leading to the diabetes occcurence.

6)Line 106 - Could you explain "GLUT" abbreviation?

The article requires major improvement. An article is not clear and coherent. The structure of the research work has not been presented. 

English language is understandable - minor editing is required (for example a repetition of "due to" - line 30, a word "namely", style, grammar structures).

Round 2

Reviewer 1 Report

Now the article is fine for me.

Reviewer 2 Report

The Authors have included all suggestions. The study has improved.